stigma; mental health; rural; urban; Bangladesh

**Corresponding author:**
Md. Omar Faruk;
Email: orhaanfaruk07@gmail.com

# Mental illness stigma in Bangladesh: Findings from a cross-sectional survey

Md Omar Faruk[1] , Abid Hasan Khan[2], Kamal Uddin Ahmed Chowdhury[1,3] , Sabiha Jahan[1], Depon Chandra Sarker[4], Erminia Colucci[5] and M. Tasdik Hasan[6,7]

[1]Department of Clinical Psychology, University of Dhaka, Dhaka, Bangladesh; [2]Department of Public Health and Informatics, Bangabandhu Sheikh Mujib Medical University, Dhaka, Bangladesh; [3]Nasirullah Psychotherapy Unit, Department of Clinical Psychology, University of Dhaka, Dhaka, Bangladesh; [4]Child Development Center, Department of Pediatrics, Satkhira Medical College and Hospital, Satkhira, Bangladesh; [5]Department of Psychology, Middlesex University, London, UK; [6]Action Lab, Department of Human Centred Computing, Faculty of Information Technology, Monash University, Melbourne, VIC, Australia and [7]Department of Public Health, State University of Bangladesh, Dhaka, Bangladesh

## Abstract

**Background:** Mental illness stigma is universally prevalent and a significant barrier to achieving global mental health goals. Mental illness stigma in Bangladesh has gained little attention despite its widespread impact on seeking mental health care in rural and urban areas. This study aimed to investigate mental illness stigma and the associated factors in rural and urban areas of Bangladesh.
**Methods:** The study areas were divided into several clusters from which 325 participants (≥18 years) were recruited with systematic random sampling. The Bangla version of the Days' Mental Illness Stigma Scale was used to collect data. Independent-samples *t*-test, ANOVA, and multiple regression were performed.
**Results:** Results suggest that gender, age, geographical location, socioeconomic status, and occupation significantly differed across subscales of stigma. Age, gender, seeking treatment of mental illness, having knowledge on mental health, and socioeconomic status were predictive factors of mental illness stigma. The results also showed a high treatment gap in both rural and urban areas.
**Conclusion:** This study supports that mental illness stigma is prevalent in Bangladesh, requiring coordinated efforts. Results can inform the development of contextually tailored mental health strategies to reduce stigma and contribute to the promotion of mental health of individuals and communities across Bangladesh.

## Impact statement

Amid the knowledge gap in the existing literature, our study provides novel insights into the mental illness stigma across geographical locations in Bangladesh. The study employed cluster sampling which reduces the potential bias in recruiting participants across study sites. A number of sociodemographic variables such as gender, age, location, socioeconomic status, history of seeking mental health care, and prior knowledge about mental health were significantly differed and associated with different aspects of mental illness stigma (i.e., anxiety, relationship disruption, hygiene, visibility, treatability, professional efficiency, and recovery). Low- and middle-income countries (LMICs), in general, witness a high treatment gap and our study supports this notion with findings suggesting a high treatment gap in both rural and urban settings. This treatment gap warrants immediate steps to be undertaken in order to reduce the global burden caused by mental illness taking the prevailing stigma into account. We hope the study facilitates more research into mental illness stigma in LMICs including Bangladesh. Furthermore, the findings can inform policy development, influence regulations, or contribute to evidence-based decision-making. The findings can also be utilized to spark a dialogue about the actions needed to shape policies or practices at various levels, from local to national or international contexts.

## Introduction

Despite the significant improvement in visibility of mental health care across the world over the last decades, many people still refrain from seeking treatment or continuing the care (Corrigan, 2004; Ciftci, 2012; Tan et al., 2020). Potential reasons include negative attitude toward mental health care, the lack of knowledge and awareness, difficulties in accessing care, particularly in rural areas, and dissatisfaction with care (i.e., biomedical model and the frequent use of

traditional and faith-based healers; Ahmedani, 2011; Green and Colucci, 2020). However, stigma has been indicated as one of the most significant factors responsible for the perpetuation of avoiding mental health care across the world (Sayers, 2001; Ciftci, 2012; Sickel et al., 2014). Mental illness stigma refers to the disgrace, social disapproval, or social discrediting of people with mental health problems (Goffman, 2009). At the heart of stigma are the elements of labeling, stereotyping, prejudice, rejection, social isolation, status loss, ignorance, low self-esteem, low self-efficacy, discrimination, and marginalization (Corrigan et al., 2006; Ahmedani, 2011). Mental illness stigma affects virtually all domains of life (Ahmedani, 2011; Corrigan et al., 2014; Sickel et al., 2014). For example, stigma diminishes self-esteem (Link et al., 2001; Corrigan, 2004), self-efficacy (Fung et al., 2007; Kleim et al., 2008), strips people of social benefits (Corrigan, 2004), support system, provider network, and community resources (Corrigan et al., 2014). Evidence also suggests that interpersonal relationships are adversely impacted due to mental illness stigma (Wong et al., 2009; Boyd et al., 2010; Gray et al., 2010). Increased mental health symptoms, decreased coping skills, and reduced compliance to treatment are also associated with mental illness stigma (Kingston Stevens et al., 2009; Verhaeghe et al., 2010). Stigma also causes physiological consequences such as obesity, back pain, and sexual health (Sickel et al., 2014). Considering the pervasive impact, stigma has been identified as a major social concern for people with mental health problems as well as families when it comes to receiving care even in this age of rapid emergence of mental health disorders globally (Sickel et al., 2014).

While stigma is a universal phenomenon, much of the scholarly works focused on western contexts suggesting a scarcity of research literature on stigma in developing countries (Lauber and Rössler, 2007; Waqas et al., 2014). Nevertheless, limited scholarly works conducted in the contexts of developing countries, especially in the south-Asian countries suggested the widespread presence of mental illness stigma. For example, in India, people diagnosed with schizophrenia reported higher perceived stigma alongside discriminatory behaviors demonstrated by family members and the community (Shrivastava et al., 2011). A study conducted in several countries including India and Nepal showed that stigma related to mental health was found to be a major barrier to accessing care (Petersen et al., 2017). In Pakistan, stigma was also found to be prevalent in which mental illnesses were believed to be caused by the possession of demons and magical spells cast by enemies while treatment of mental illnesses is largely depended on shamans (traditional healers) using talismans, amulets, and incantations (Waqas et al., 2014; Husain et al., 2020; Munawar et al., 2020). Mental illness stigma in other south-Asian countries such as Bhutan (Pelzang, 2012), Sri Lanka (Fernando, 2010), Bangladesh (Hasan and Thornicroft, 2018; *stigma in the context of Bangladesh is presented in the Bangladesh perspective section in detail*), and Afghanistan (Nine et al., 2022) was also found to be prevalent. In general, people with mental illness in Asia are labeled as dangerous and aggressive increasing the likelihood of maintaining social distance by people with no mental illness. There is a widespread belief that mental illnesses are influenced by magic, religion, and supernatural entities. Stigma from family members is pervasive, and families having members with mental illness face social disapproval and devaluation, particularly in the context of marriage, marital separation, and divorce. Psychiatric symptoms are viewed as socially disadvantaged compared to somatic symptoms, leading to widespread somatization of psychiatric disorders

in the region (Lauber and Rössler, 2007; Zhang et al., 2020). Mental illness stigma appears not only a public health issue but a human rights issue in low- and middle-income countries (LMICs) including south-Asian countries where people with mental illness are often subject to widespread abuse and social exclusion (Naslund and Deng, 2021). Stigma reduction strategies are also inadequate, under-funded, and under-studied in LMICs (Naslund and Deng, 2021).

It is often argued that mental health conditions and relevant services are disproportionately distributed in both high as well as LMICs (Abdulmalik and Thornicroft, 2016; Khoury and Daouk, 2017). The majority of the LMICs face a high prevalence of mental disorders, critical challenges in ensuring minimum mental health care, and acute shortage of mental health professionals (Rathod et al., 2017; Rojas et al., 2019). In addition, LMICs also witness unequal geographic distribution of care with overstressing on urban care, cultural and religious restrictions and attributes to illness, and belief systems, social, and contextual factors (e.g., poverty, internal migration, lifestyle changes, etc.) synergistically acting as the factors for the high burden of mental illnesses and low access to mental health care (Rathod et al., 2017; Rojas et al., 2019) and stigma toward mental illnesses has been identified as the significant barrier to treatment in LMICs (Thornicroft et al., 2010).

## Mental illness stigma: Bangladesh perspective

Mental illness stigma has been found to be widespread in Bangladesh that often overwhelms mental health care services (Hasan and Thornicroft, 2018). A study conducted in 2013 in the context of Bangladesh alongside western countries, showed that some concepts of depression and schizophrenia were found to be disproportionately stigmatized compared to countries in the west (Pescosolido et al., 2013). Another study suggested that disclosure spillover concerns – an idea that refers to the anticipation of negative consequences among friends and families if mental illnesses of an individual are disclosed led to the discriminatory attitudes such as social avoidance (Krendl and Pescosolido, 2020). The recent *National Mental Health Survey 2018–2019* demonstrated that people with mental health problems reported that visiting mental health professionals might result in labeling with derogatory terms such as 'mad' (World Health Organization, 2019; Hasan et al., 2021a). While access to mainstream mental health care is limited, similar to other LMICs (Rathod et al., 2017), a substantial percentage of people with mental health problems make use of a range of traditional and faith-based health care providers such as traditional and faith healers (pirs and fakirs), and homeopathic practitioners as their first point of contact for health care (Hasan et al., 2021b). Access to mental health care is inadequate and unequal in Bangladesh due to stigma and misconception around mental illness (Nuri et al., 2018). Strong levels of stigma and lack of awareness in Bangladesh has been identified as barriers to accessing mental health care (Nuri et al., 2018). Evidence suggests that the delay in seeking care for mental health illness was associated with widespread stigma (Islam et al., 2008). The stigma attached to mental health problems can have an adverse impact on help-seeking behavior with many people suffering in silence and experiencing isolation and discrimination, including human rights abuses (Abdulmalik and Thornicroft, 2016; Hasan and Thornicroft, 2018; Hasan et al., 2021a). It is reasonable to assume that persistent stigma and the resulting

discrimination may continue to hamper the access to mental health care despite the availability of mental health care (Nuri et al., 2018). Therefore, it was recommended that counteracting stigma and discrimination in Bangladesh required mental health literacy and pertinent strategies (Patel and Thornicroft, 2009; Hasan et al., 2021b). Factors associated with mental illness stigma can be of paramount importance to develop such strategies or programs in rural and urban areas of Bangladesh. However, study findings, especially the factors associated with mental illness stigma that can be utilized are limited. Against this backdrop, this present study aimed to investigate mental illness stigma among rural and urban populations in Bangladesh along with the predictive factors. The study is expected to contribute to the understanding of mental illness stigma that can inform the development of suitable strategies to reduce stigma in a country where resources are limited.

## Methods

### Setting and participants

The present study was conducted under a project led by Nasirullah Psychotherapy Unit in association with ADD International Bangladesh with the aim of implementing 'Community-based Mental Health Project' in rural and urban areas of Bangladesh. The South-Western regions namely Bastali union, adjacent villages of Bagerhat Sadar Upazila, Gotapara union, and Rampal Upazila comprised the rural areas. An adjacent area in Jashore, a South-Western part of Bangladesh, was also included in the rural areas. Urban areas included different locations in Dhaka city such as Khilgoan, Badda, and Bauniabadh. The urban and rural areas were divided into clusters according to ward number. A list of wards was obtained from the respective organizations for persons with disabilities (OPD) and organizations partnered with the project. The wards were used to form clusters using a specific range (every two ward formed a cluster). We double-checked the selected clusters to ensure each ward was included representing diversity and characteristics of the population within each cluster. We assessed the homogeneity of each cluster in terms of relevant demographics to determine if they adequately represented the source population. We found no significant differences. The homogeneity of selected clusters can be attributed to the geographical location (urban and rural in this case) where residents share similar characteristics in terms of income and socioeconomic status (SES). The clusters were then randomly selected for data collection. Random selection of clusters helps ensure each cluster has an equal chance of being included in the sample, thereby reducing the potential for selection bias. After the clusters were determined, we employed systematic random sampling to collect data from the selected clusters. Comparing the characteristics of the sample to the characteristics of the total population of the country involves conducting a process called sample-to-population comparison. This analysis is used to determine whether the sample is representative of the larger population and to identify any differences between the two. We recruited participants with systematic random sampling in order to minimize selection bias – an approach to sample-to-population comparison. However, there are other ways that could have been embraced to compare the characteristics (i.e., running statistical tests such as chi-square for categorical variables or *t*-tests or ANOVA for continuous variables to determine if there are significant differences) after collecting data from both the sample and the total population. A total of 350 adult participants (≥18 years) were recruited from rural and urban settings. However, 325 data were retained for data analysis after the removal of incomplete data. Male participants (67.9%) outnumbered female (31.5%) participants aged between 18 and 80.

### Procedures

A total of 20 research assistants collected the data after training provided by the principal researcher (M.O.F.). The research assistants were paid volunteers. The concept of stigma and its association with mental health, administration of the scales, and cognitive interviewing were included in the structured training module. Each area was divided into several wards (clusters) with a corresponding number. Participants were systematically recruited (every three families with one adult male or female) from each ward. The diagnosis of mental disorders followed a structured interview in terms of comprehensive assessment. This included a comprehensive range of diagnostic criteria and symptom domains such as mood, anxiety, psychotic symptoms, and substance use to determine the presence or absence of mental disorders. The structured interview covered the specific criteria outlined in the diagnostic manual such as Diagnostic and Statistical Manual (DSM-5; *Diagnostic and Statistical Manual of Mental Disorders*, 2013). Participants with intellectual disabilities and those diagnosed with severe mental health illnesses (e.g., schizophrenia and bipolar mood disorder) were excluded from the study. The exclusions were made by the research assistants in accordance with the DSM-5 criteria. Data were collected at a time when the country-wide lockdown (due to the COVID-19 pandemic) was lifted. Nevertheless, adequate safety measures were followed during the data collection. A verbal and written consent form was given to the participants where the nature and purpose of the study were explicitly mentioned. In addition, the implications of the study and the right to withdraw from taking part in the study were also described. Participants with no literacy were assisted by research assistants. A thumb mark was used to indicate consent for those with little or no literacy at all. Participants were given a referral directory of available mental health services across the country as part of the safety protocol. No one refused to take part in the study. Potential reasons may include cultural norms and expectations that influence participants to participate in a study, particularly in close-knit communities. Participants may have perceived potential benefits from participating in the study such as access to information or resources that could be helpful to them (a list of mental health services across the country was provided). In addition, participants may have a personal interest in the study topic or may have found the study relevant to their experiences contributing to their motivation in taking part in the study. Finally, participants may have felt a sense of altruism or a desire to contribute to the advancement of knowledge or help others by participating in the study. The participation was voluntary, and no monetary compensation was provided.

## Measures

### Sociodemographic measures

The demographic measures included age, sex, geographical location, educational status, occupation, SES, religion, treatment sought for any types of mental health problems, relationship status, monthly income, number of family members, knowledge about

mental health (yes/no), and presence of family members with mental health problems (yes/no).

### The Mental Illness Stigma Scale

The Mental Illness Stigma Scale was developed by Day et al. (2007). The Bangla translated version was used in the National Mental Health Survey (World Health Organization, 2019) conducted by the National Institute of Mental Health and Hospital in association with the World Health Organization (WHO). The scale was translated and back-translated by the hospital authority following the suggested procedures of scale adaptation. Besides, the scale was pretested on a group of 300 people prior to the final administration of the survey. However, no published data on the psychometric properties of the Bangla version of the scale is available. The 28-item Likert-type scale measures seven factors of attitudes toward people with mental illness: interpersonal anxiety, relationship disruption, poor hygiene, visibility, treatability, professional efficacy, and recovery. The scale was designed to measure the attitudes toward people with mental illness, depression, bipolar mood disorder, and schizophrenia (Day et al., 2007). The scale was used in a variety of studies (see Masuda et al., 2009a, b; Michalak et al., 2014; Young et al., 2019). The scale was used in the study amid the unavailability of culturally validated scales that assess mental illness stigma in Bangladesh. The scale was deemed suitable as the WHO and NIMH used it to conduct the nationwide mental health survey in Bangladesh. Prior to the use in the present study, the scale was administered to a sample of 50 participants (excluding the main sample) in both rural and urban areas to understand the comprehensibility of items. No item was found to be difficult to understand, therefore, no change was required. The Cronbach's alpha for the total scale in the present study was 0.67. The Cronbach's alphas for the subscales in the present study were 0.71, 0.50, 0.52, 0.76, 0.46, 0.52, and 0.74, respectively. We assessed test–retest reliability of the scale on a sample of an additional 100 participants in both settings with a gap of 2 weeks. The test–retest reliability coefficient for the present study was found to be $r = 0.81$ at $p < 0.01$.

### Statistical analysis

SPSS version 25.0 was used to analyze the demographic information (frequencies, percentages, and means) and to compute statistical tests (e.g., independent sample *t*-tests and ANOVA). Multiple regression was used to assess significant variables predictive of stigma toward mental illness. The decision to report both ANOVA and regression was made to investigate the research objectives. The research questions involved group comparisons or variance analysis; therefore, ANOVA was thought to provide additional insights. We used ANOVA to test differences among multiple groups whereas regression was used to examine the relationship between variables. Findings of multiple regression were construed with 95% confidence intervals.

### Results

About three-fourths of the participants were aged 25 or above (75.7%). The mean age of the participants was 37.08 (SD = 14.8). Additionally, more than half of the participants were from outside of Dhaka (53.8%). Majority of the participants were male (68.3%), married (56.9%) and Muslim (80.9%). About one-fourth of the participants were businessperson (25.8%) and most of the

participants belonged to middle (37.8%) or lower–middle (35.5%) SES. Furthermore, about 26% of the participants were educated up to higher secondary level (HSC) and 6.2% of the participants were illiterate or had no formal education. It was also found that about 78% of the participants were aware or had knowledge about mental health. Meanwhile, only 2% of the participants had family history of mental health illness and overall, 4% of participants sought for mental illness treatment (Table 1).

Mental illness stigma scores were reported in accordance with the subscales – anxiety (*M* = 33.37; SD = 6.832), relationship

**Table 1.** Demographic properties of participants (*N* = 325)

| Participants characteristics | *N* (%) |
|---|---|
| Age | *M* = 37.08; SD = 14.82 |
| <25 | 79 (24.3%) |
| 25–40 | 122 (37.5%) |
| >40 | 124 (38.2%) |
| Location | |
| Dhaka | 150 (46.2%) |
| Outside Dhaka | 175 (53.8%) |
| Gender | |
| Male | 222 (68.3%) |
| Female | 103 (31.7%) |
| Socioeconomic status (SES) | |
| Lower SES | 74 (22.6%) |
| Lower–middle SES | 116 (35.5%) |
| Middle SES | 123 (37.8%) |
| Higher SES | 12 (3.7%) |
| Occupation | |
| Student | 79 (24.3%) |
| Service holder | 67 (20.6%) |
| Businessperson | 84 (25.8%) |
| Housewife | 54 (16.6%) |
| Unemployed | 41 (12.6%) |
| Marital status | |
| Unmarried | 117 (36%) |
| Married | 185 (56.9%) |
| Widow/widower | 23 (7.1%) |
| Religion | |
| Muslim | 263 (80.9%) |
| Non-Muslim | 62 (19.1%) |
| Literacy | |
| Up to primary | 40 (12.3%) |
| SSC | 71 (21.8%) |
| HSC | 83 (25.5%) |
| Honors | 60 (18.5%) |
| Master's and above | 51 (15.7%) |

*(Continued)*

**Table 1.** (*Continued*)

| Participants characteristics | N (%) |
|---|---|
| Illiterate | 20 (6.2%) |
| Knowledge about mental health | |
| Yes | 253 (77.8%) |
| No | 72 (22.2%) |
| Presence of mental illness among family members | |
| Yes | 7 (2.2%) |
| No | 318 (97.8%) |
| Treatment sought for mental health illness | |
| Yes | 13 (4.0) |
| No | 312 (96.0) |

Abbreviations: HSC, higher secondary school certificate; SSC, secondary school certificate.

disruption ($M = 27.64$; SD = 5.574), hygiene ($M = 20.88$; SD = 3.819), visibility ($M = 15.32$; SD = 5.227), treatability ($M = 15.98$; SD = 3.402), professional efficiency ($M = 11.32$; SD = 2.424), and recovery ($M = 11.04$; SD = 2.853; Table 2).

The study further showed that there was significant mean difference of multiple subscale scores for mental illness stigma (anxiety: $F(2,322) = 15.5$, *p*-value < 0.01; relationship disruption: $F(2,322) = 9.2$, *p*-value < 0.01; visibility: $F(2,322) = 7.2$, *p*-value < 0.01) among the age groups (Table 3). Dunnett post hoc analysis showed that the mean difference of mentioned subscale scores was significantly different between participants aged less than 25 and participants aged more than 40 as well as participants aged 25–40 and participants age more than 40 years, however, there was no statistical significance in terms of association between participants aged below 25 and participants aged 25–40 years (Table 4).

Addition to that, the study found that means of multiple subscale scores (relationship disruption: $F(1,323) = 13.8$, *p*-value < 0.01; hygiene: $F(1,323) = 4.9$, *p*-value < 0.01; visibility: $F(1,323) = 21.6$, *p*-value < 0.01; treatability: $F(1,323) = 6.5$, *p*-value < 0.01; recovery: $F(1,323) = 10.9$, *p*-value < 0.01) were significantly different between participants residing within Dhaka and participants residing outside Dhaka (Table 3). It was also noticed that mean scores of anxiety subscale ($F(1,323) = 6.18$, *p*-value < 0.01), relationship disruption subscale ($F(1,323) = 6.3$, *p*-value < 0.01) and hygiene subscale ($F(1,323) = 8.9$, *p*-value < 0.01) differed by gender (Table 3).

Furthermore, the study showed that difference in mean scores was statistically significant in terms of all the subscales except for

**Table 2.** Means scores of mental illness stigma sub-scales

| Stigma subscales (with range) | Mean | SD |
|---|---|---|
| Anxiety (7–49) | 33.37 | 6.832 |
| Relationship disruption (6–42) | 27.64 | 5.574 |
| Hygiene (4–28) | 20.88 | 3.819 |
| Visibility (4–28) | 15.32 | 5.227 |
| Treatability (3–21) | 15.98 | 3.402 |
| Professional efficiency (2–14) | 11.32 | 2.424 |
| Recovery (2–14) | 11.04 | 2.853 |

visibility subscale (anxiety: $F(3,321) = 4.54$, *p*-value < 0.01; relationship disruption: $F(3,321) = 4.3$, *p*-value < 0.01; hygiene: $F(3,321) = 10.8$, *p*-value < 0.01; treatability: $F(3,321) = 16.1$, *p*-value < 0.01; professional efficiency: $F(3,321) = 3.8$, *p*-value < 0.05; recovery: $F(3,321) = 5.8$, *p*-value < 0.01) among the participants of different SES (Table 3). Post hoc analysis revealed that the mean score difference was significant for lower SES, lower–middle SES (treatability, PE, and recovery subscales) and middle SES (anxiety, RD, hygiene, and treatability subscales). However, there was no significant mean subscale score difference between higher SES and other SES groups (Table 4). Moreover, participants' educational status, occupation, knowledge of mental health, presence of mental illness among family members and treatment sought for mental health illness showed significant mean score differences in terms of at least one of seven subscales of mental illness stigma scale (Table 3).

Multiple regression analysis showed that participants aged greater than 40 were likely to have increased score in anxiety subscale ($\beta = 4.64$; 95% CI: 1.33 to 7.96) compared to participants aged below 25. Participants residing outside Dhaka were likely to have increased scores in RD subscale ($\beta = 1.79$; 95% CI: 0.57 to 3.00) but decreased scores in visibility ($\beta = -2.47$; 95% CI: $-3.64$ to $-1.31$) and recovery subscales ($\beta = -0.93$; 95% CI: $-1.57$ to $-0.30$). It was further seen that female participants were likely to have increased scores for anxiety ($\beta = 2.21$; 95% CI: 0.05 to 4.36), RD ($\beta = 2.22$; 95% CI: 0.49 to 3.96) and hygiene subscales ($\beta = 1.44$; 95% CI: 0.23 to 2.65) compared to their male counterpart. Additionally, participants of lower–middle SES were likely to have increased score for treatability subscale ($\beta = 1.61$; 95% CI: 0.53 to 2.68) in reference to participants of lower SES. Moreover, having knowledge of mental health was significantly associated with increased scores for hygiene ($\beta = 1.18$; 95% CI: 0.01 to 2.34), visibility ($\beta = 2.87$; 95% CI: 0.59 to 3.78) and professional efficiency subscales ($\beta = 1.46$; 95% CI: 0.73 to 2.20) compared to participants with no or not enough knowledge of mental health. However, participants seeking treatment for mental health illness were likely to have decreased score for PE subscale ($\beta = -1.85$; 95% CI: $-3.18$ to $-0.52$) compared to participants not seeking any treatment for mental health illness (Table 5).

## Discussion

The present cross-sectional study was conducted against the backdrop of the limited evidence available on mental illness stigma in rural and urban areas of Bangladesh and investigated the mental illness stigma among adult population.

The results suggest that females reported more mental illness stigma in terms of anxiety, relationship disruption and hygiene subscales than their male counterparts. For the remaining subscales (e.g., visibility, treatability, professional efficiency, and recovery), no significant statistical difference was observed by gender. Previous studies showed that females tend to report more stigma than males (Farina, 1981; Anderson et al., 2015; Schroeder et al., 2021) implying an association between gender and mental illness stigma. However, more recent studies have produced results showing that females report less stigmatizing attitudes toward mental illness than males (Wirth and Bodenhausen, 2009; Bradbury, 2020; Conceição et al., 2022). In addition, the nationwide survey in Bangladesh found that there is no significant difference in stigma between males and females (World Health Organization, 2019). However, the present study produced counterproductive results about gender

**Table 3.** Analysis of variance (ANOVA) of sociodemographic and subscales of mental illness stigma scale ($n = 325$)

| Variables | Anxiety M (SD) | Anxiety F (df) | RD M (SD) | RD F (df) | Hygiene M (SD) | Hygiene F (df) | Visibility M (SD) | Visibility F (df) | Treatability M (SD) | Treatability F (df) | PE M (SD) | PE F (df) | Recovery M (SD) | Recovery F (df) |
|---|---|---|---|---|---|---|---|---|---|---|---|---|---|---|
| **Age** | | | | | | | | | | | | | | |
| <25 | 30.66 (7.3) | 15.5** (2,322) | 26.08 (5.9) | 9.2** (2,322) | 20.03 (4.2) | 3.0 (2, 322) | 16.91 (5.5) | 7.2** (2,322) | 16.44 (3.4) | 1.1 (2, 322) | 11.10 (2.8) | 0.83 (2, 322) | 11.39 (2.8) | 1.5 (2,322) |
| 25–40 | 32.71 (6.1) | | 27.06 (5.6) | | 20.95 (4.0) | | 15.50 (5.2) | | 15.89 (3.8) | | 11.53 (2.5) | | 10,70 (3.2) | |
| >40 | 35.73 (6.5) | | 29.22 (4.9) | | 21.35 (3.3) | | 14.14 (4.9) | | 15.77 (2.9) | | 11.26 (2.1) | | 11.15 (2.5) | |
| **Location** | | | | | | | | | | | | | | |
| Dhaka | 32.76 (7.8) | 2.20 (1,323) | 26.43 (6.3) | 13.8** (1, 323) | 21.39 (4.1) | 4.9* (1, 323) | 16.73 (5.7) | 21.6** (1, 323) | 16.49 (3.5) | 6.5* (1, 323) | 11.54 (2.6) | 2.2 (1, 323) | 11.50 (2.8) | 10.9** (1, 323) |
| Outside Dhaka | 33.89 (5.8) | | 28.69 (4.6) | | 20.45 (3.5) | | 14.11 (4.5) | | 15.54 (3.3) | | 11.14 (2.3) | | 10.57 (2.8) | |
| **Gender** | | | | | | | | | | | | | | |
| Male | 32.73 (6.7) | 6.18* (1, 323) | 27.12 (5.5) | 6.3* (1,323) | 20.45 (3.7) | 8.9** (1, 323) | 15.60 (5.4) | 2.0 (1, 323) | 15.87 (3.4) | 0.6 6 (1, 323) | 11.35 (2.4) | 0.1 (1, 323) | 11.09 (2.8) | 0.2 (1, 323) |
| Female | 34.74 (6.8) | | 28.78 (5.6) | | 21.80 (3.9) | | 14.73 (4.8) | | 16.20 (3.5) | | 11.27 (2.3) | | 10.93 (3.0) | |
| **Socioeconomic status (SES)** | | | | | | | | | | | | | | |
| Lower SES | 35.47 (6.9) | 4.54** (3, 321) | 29.42 (4.8) | 4.3** (3, 321) | 21.81 (3.2) | 10.8** (3, 321) | 14.38 (5.1) | 1.6 (3, 321) | 14.09 (2.8) | 16.1** (3, 321) | 10.55 (2.2) | 3.8* (3, 321) | 10.12 (2.5) | 5.8** (3, 321) |
| Lower–middle SES | 33.28 (7.4) | | 27.67 (6.0) | | 21.87 (3.8) | | 15.93 (5.8) | | 17.35 (2.6) | | 11.66 (2.2) | | 11.79 (2.07 | |
| Middle SES | 31.98 (6.1) | | 26.53 (5.5) | | 19.46 (3.9) | | 15.44 (4.8) | | 15.77 (3.7) | | 11.39 (2.7) | | 10.94 (3.04) | |
| Higher SES | 35.42 (3.9) | | 27.83 (4.1) | | 20.17 (2.8) | | 14.08 (4.0) | | 16.42 (4.1) | | 12.08 (1.9) | | 10.50 (3.2) | |
| **Occupation** | | | | | | | | | | | | | | |
| Student | 31.32 (7.1) | 4.36** (4, 320) | 26.33 (6.2) | 3.9** (4, 320) | 20.16 (4.3) | 2.1 (4, 320) | 16.96 (5.5) | 2.9* (4, 320) | 16.58 (3.4) | 1.6 (4, 320) | 11.25 (2.8) | 1.4 (4, 320) | 11.58 (2.7) | 1.6 (4, 320) |
| Service holder | 33.13 (5.6) | | 27.58 (5.4) | | 20.69 (4.0) | | 14.91 (4.9) | | 16.33 (3.7) | | 11.63 (2.5) | | 11.07 (2.8) | |
| Businessperson | 34.52 (6.6) | | 28.79 (4.7) | | 20.98 (3.4) | | 14.81 (5.0) | | 15.67 (2.9) | | 11.63 (2.0) | | 11.08 (2.8) | |
| Housewife | 35.67 (7.0) | | 29.09 (5.5) | | 22.06 (3.7) | | 14.28 (4.7) | | 15.30 (3.4) | | 10.91 (2.2) | | 10.39 (3.0) | |
| Unemployed | 32.29 (7.2) | | 26.02 (5.6) | | 20.83 (3.4) | | 15.27 (5.9) | | 15.78 (3.8) | | 10.88 (2.6) | | 10.73 (2.9) | |
| **Literacy** | | | | | | | | | | | | | | |
| Up to primary | 32.48 (9.2) | 3.74** (5, 319) | 26.75 (6.5) | 4.8** (5, 319) | 22.10 (3.6) | 2.4* (5, 319) | 15.52 (5.7) | 2.5* (5, 319) | 14.72 (3.6) | 7.6** (5, 319) | 10.90 (2.2) | 4.0** (5, 319) | 9.95 (3.0) | 3.8** (5, 319) |
| SSC | 34.45 (5.8) | | 29.08 (4.5) | | 21.44 (3.6) | | 15.10 (5.2) | | 15.45 (2.6) | | 11.25 (2.1) | | 11.08 (2.4) | |
| HSC | 33.19 (7.3) | | 28.20 (5.4) | | 20.22 (3.9) | | 15.31 (4.5) | | 16.28 (3.0) | | 11.17 (2.6) | | 11.20 (3.0) | |

(*Continued*)

| Variables | Anxiety M (SD) | Anxiety F (df) | RD M (SD) | RD F (df) | Hygiene M (SD) | Hygiene F (df) | Visibility M (SD) | Visibility F (df) | Treatability M (SD) | Treatability F (df) | PE M (SD) | PE F (df) | Recovery M (SD) | Recovery F (df) |
|---|---|---|---|---|---|---|---|---|---|---|---|---|---|---|
| Honors | 32.02 (6.6) | | 25.60 (5.7) | | 20.20 (4.2) | | 16.85 (5.9) | | 16.88 (3.8) | | 11.77 (2.7) | | 11.38 (3.1) | |
| Masters and above | 32.35 (5.1) | | 26.61 (5.7) | | 20.61 (3.8) | | 14.86 (5.3) | | 17.29 (3.8) | | 12.14 (2.3) | | 11.84 (2.6) | |
| Illiterate | 38.65 (4.6) | | 30.75 (5.6) | | 21.95 (2.5) | | 12.35 (5.2) | | 13.05 (2.03) | | 9.64 (1.9) | | 9.35 (2.1) | |
| Religion | | | | | | | | | | | | | | |
| Muslim | 33.43 (6.7) | 0.11 (1, 323) | 27.69 (5.5) | 0.1 (1, 323) | 20.92 (4.1) | 0.13 (1, 323) | 15.53 (5.3) | 2.2 (1,323) | 16,02 (3.5) | 0.2 (1, 323) | 11.42 (2.4) | 2.3 (1, 323) | 10.94 (2.9) | 2.0 (1, 323) |
| Non-Muslim | 33.11 (7.5) | | 27.45 (6.0) | | 20.73 (2.4) | | 14.45 (4.8) | | 15.82 (2.8) | | 10.90 (2.4) | | 11.50 (2.5) | |
| Knowledge about mental health | | | | | | | | | | | | | | |
| Yes | 32.81 (6.8) | 7.61** (1, 323) | 27.34 (5.6) | 3.4 (1, 323) | 21.01 (4.0) | 1.3 (1, 323) | 15.89 (5.5) | 13.9** (1, 323) | 16.64 (3.2) | 49.7** (1, 323) | 11.73 (2.3) | 35.1** (1, 323) | 11.33 (2.9) | 12.1** (1, 323) |
| No | 35.31 (6.8) | | 28.71 (5.3) | | 20.43 (3.0) | | 13.33 (3.7) | | 13.65 (2.9) | | 9.90 (2.2) | | 10.03 (2.4) | |
| Presence of mental illness among family members | | | | | | | | | | | | | | |
| Yes | 34 (6.8) | 0.06 (1, 323) | 29.29 (5.4) | 0.6 (1, 323) | 18.0 (4.6) | 4.1* (1, 323) | 17.43 (3.9) | 1.2 (1, 323) | 13.71 (3.0) | 3.2 (1, 323) | 12.00 (2.00) | 0.6 (1, 323) | 11.14 (3.2) | 0.01 (1, 323) |
| No | 33.35 (6.8) | | 27.61 (5.9) | | 20.94 (3.9) | | 15.28 (5.2) | | 16.03 (3.4) | | 11.31 (2.4) | | 11.04 (2.9) | |
| Treatment sought for mental health illness | | | | | | | | | | | | | | |
| Yes | 30.23 (4.8) | 2.87 (1, 323) | 27.08 (4.9) | 0.14 (1,323) | 20.31 (6.1) | 0.3 (1, 323) | 16.08 (6.0) | 0.3 (1, 323) | 15.69 (4.2) | 0.1 (1, 323) | 10.00 (2.3) | 4.1* (1, 323) | 11.15 (3.0) | 0.02 (1, 323) |
| No | 33.50 (6.9) | | 27.67 (5.6) | | 20.90 (3.7) | | 15.29 (5.2) | | 15.99 (3.4) | | 11.38 (2.4) | | 11.04 (2.9) | |

Abbreviations: df, degrees of freedom; M, mean; PE, professional efficiency; RD, relationship disruption; SD, standard deviation.
*$p$-value<0.05.
**$p$-value<0.01.

**Table 4.** Post hoc analysis of sociodemographic variables and subscales of mental illness stigma scale ($n$ = 325)

| Variables | | Anxiety MD (95% CI) | RD MD (95% CI) | Hygiene MD (95% CI) | Visibility MD (95% CI) | Treatability MD (95% CI) | PE MD (95% CI) | Recovery MD (95% CI) |
|---|---|---|---|---|---|---|---|---|
| Age | | | | | | | | |
| <25 | 25–40 | −2.055 | −0.981 | −0.926 | 1.411 | 0.558 | −0.432 | 0.687 |
| | | (−4.45 to 0.34) | (−3.01 to 1.05) | (−2.36 to 0.51) | (−0.46 to 3.28) | (−0.69 to 1.81) | (−1.38 to 0.51) | (−0.34 to 1.72) |
| | >40 | −5.076** | −3.142** | −1.33 | 2.774** | 0.669 | −0.157 | 0.239 |
| | | (−7.51 to −2.65) | (−5.07 to −1.21) | (−2.67 to 0.01) | (0.95 to 4.60) | (−0.45 to 1.79) | (−1.05 to 0.73) | (−0.69 to 1.17) |
| 25–40 | <25 | 2.055 | 0.981 | 0.926 | −1.411 | −0.558 | 0.432 | −0.687 |
| | | (0.34 to −4.45) | (−1.05 to 3.01) | (−.51 to 2.36) | (−3.28 to 0.46) | (−1.81 to 0.69) | (−.51 to 1.38) | (−1.72 to 0.34) |
| | >40 | −3.021** | −2.160** | −0.404 | 1.363 | 0.111 | 0.275 | −0.448 |
| | | (−4.94 to −1.10) | (−3.78 to −0.54) | (−1.53 to 0.72) | (−0.17 to 2.90) | (−0.93 to 1.15) | (−0.42 to 0.97) | (−1.33 to 0.43) |
| >40 | <25 | 5.076** | 3.142** | 1.33 | −2.774** | −0.669 | 0.157 | −0.239 |
| | | (2.65 to 7.51) | (1.21 to 5.07) | (−.01 to 2.67) | (−4.60 to −0.95) | (−1.79 to 0.45) | (−0.73 to 1.05) | (−1.17 to 0.69) |
| | 25–40 | 3.021** | 2.160** | 0.404 | −1.363 | −0.111 | −0.275 | 0.448 |
| | | (1.10 to 4.94) | (0.54 to 3.78) | (−.72 to 1.53) | (−2.90 to 0.17) | (−1.15 to 0.93) | (−0.97 to 0.42) | (−0.43 to 1.33) |
| Socioeconomic status (SES) | | | | | | | | |
| Lower SES | Lower–middle SES | 2.197 | 1.747 | −0.06 | −1.553 | −3.259** | −1.110** | −1.671** |
| | | (−.61 to 5.00) | (−.34 to 3.83) | (−1.41 to 1.29) | (−3.66 to 0.56) | (−4.35 to −2.17) | (−1.99 to −0.23) | (−2.68 to −0.66) |
| | Middle SES | 3.489** | 2.890** | 2.356** | −1.061 | −1.678** | −0.836 | −0.821 |
| | | (.89 to 8.09) | (.91 to 4.87) | (1.00 to 3.71) | (−3.00 to 0.88) | (−2.93 to −0.43) | (−1.77 to 0.10) | (−1.88 to 0.24) |
| | Higher SES | 0.056 | 1.586 | 1.644 | 0.295 | −2.322 | −1.529 | −0.378 |
| | | (−3.88 to 3.99) | (−2.30 to 5.47) | (−1.01 to 4.30) | (−3.52 to 4.11) | (−6.13 to 1.48) | (−3.35 to 0.29) | (−3.33 to 2.58) |
| Lower–middle SES | Lower SES | −2.197 | −1.747 | 0.06 | 1.553 | 3.259** | 1.110** | 1.671** |
| | | (−5.00 to 0.61) | (−3.83 to 0.34) | (−1.29 to1.41) | (−0.56 to 3.66) | (2.17 to 4.35) | (0.23 to 1.99) | (0.66 to 2.68) |
| | Middle SES | 1.292 | 1.144 | 2.415** | 0.492 | 1.581** | 0.274 | 0.85 |
| | | (−1.05 to 3.63) | (−.83 to 3.12) | (1.10 to 3.73) | (−1.34 to 2.33) | (0.48 to 2.68) | (−0.57 to 1.12) | (−0.13 to 1.83) |
| | Higher SES | −2.141 | −0.161 | 1.704 | 1.848 | 0.937 | −0.42 | 1.293 |
| | | (−5.95 to 1.67) | (−4.04 to 3.72) | (−0.94 to 4.35) | (−1.94 to 5.63) | (−2.84 to 4.72) | (−2.21 to 1.37) | (−1.65 to 4.23) |
| Middle SES | Lower SES | −3.489** | −2.890* | −2.356** | 1.061 | 1.678** | 0.836 | 0.821 |
| | | (−6.09 to −0.89) | (−4.87 to −0.91)* | (−3.71 to −1.00) | (−0.88 to 3.00) | (0.43 to 2.93) | (−0.10 to 1.77) | (−0.24 to 1.88) |
| | Lower–middle SES | −1.292 | −1.144 | −2.415** | −0.492 | −1.581** | −0.274 | −0.85 |
| | | (−3.63 to 1.05) | (−3.12 to 0.83) | (−3.73 to −1.10) | (−2.33 to 1.34) | (−2.68 to −0.48) | (−1.12 to 0.57) | (−1.83 to 0.13) |

(*Continued*)

**Table 4.** (*Continued*)

| Variables | | Anxiety MD (95% CI) | RD MD (95% CI) | Hygiene MD (95% CI) | Visibility MD (95% CI) | Treatability MD (95% CI) | PE MD (95% CI) | Recovery MD (95% CI) |
|---|---|---|---|---|---|---|---|---|
| | Higher SES | −3.433 (−7.13 to 0.27) | −1.305 (−5.15 to 2.54) | −0.711 (−3.36 to 1.94) | 1.356 (−2.37 to 5.08) | −0.644 (−4.45 to 3.16) | −0.693 (−2.50 to 1.12) | 0.443 (−2.51 to 3.59) |
| Higher SES | Lower SES | −0.056 (−3.99 to 3.88) | −1.586 (−5.47 to 2.30) | −1.644 (−4.30 to 1.01) | −0.295 (−4.11 to 3.52) | 2.322 (−1.48 to 6.13) | 1.529 (−0.29 to 3.35) | 0.378 (−2.58 to 3.33) |
| | Lower–middle SES | 2.141 (−1.67 to 5.95) | 0.161 (−3.72 to 4.04) | −1.704 (−4.35 to 0.94) | −1.848 (−5.63 to 1.94) | −0.937 (−4.72 to 2.84) | 0.42 (−1.37 to 2.21) | −1.293 (−4.23 to 1.65) |
| | Middle SES | 3.433 (−.27 to 7.13) | 1.305 (−2.54 to 5.15) | 0.711 (−1.94 to 3.38) | −1.356 (−5.08 to 2.37) | 0.644 (−3.16 to 4.45) | 0.693 (−1.12 to 2.50) | −0.443 (−3.39 to 2.51) |
| Education | | | | | | | | |
| Up to primary | Up to SSC | −1.976 (−6.88 To 2.92) | −2.335 (−5.86 to 1.19) | 0.663 (−1.47 to 2.79) | 0.426 (−2.88 to 3.73) | −0.726 (−2.68 to 1.23) | −0.354 (−1.65 to 0.94) | −1.135 (−2.82 to 0.55) |
| | Up to HSC | −0.718 (−5.74 to 4.31) | −1.455 (−5.06 to 2.15) | 1.883 (−.26 to 4.02) | 0.212 (−2.91 to 3.34) | −1.552 (−3.54 to 0.44) | −0.269 (−1.62 to 1.09) | −1.255 (−3.00 to 0.49) |
| | Honors | 0.458 (−4.65 to 5.56) | 1.15 (−2.67 to 4.97) | 1.9 (−.47 to 4.27) | −1.325 (−4.88 to 2.23) | −2.158 (−4.41 to 0.09) | −0.867 (−2.34 to 0.61) | −1.433 (−3.31 to 0.45) |
| | MS and above | 0.122 (−4.80 to 5.04) | 0.142 (−3.78 to 4.07) | 1.492 (−.85 to 3.84) | 0.662 (−2.85 to 4.18) | −2.569* (−4.91 to −0.23) | −1.237 (−2.68 to 0.20) | −1.893* (−3.72 to −0.07) |
| | Illiterate | −6.175* (−11.59 to −0.76) | −4 (−8.34 to 0.34) | 0.15 (−2.30 to 2.60) | 3.175 (−.38 to 6.73) | 1.675 (−.54 to 3.89) | 1.25 (−.47 to 2.97) | 0.6 (−1.45 to 2.65) |
| Up to SSC | Up to primary | 1.976 (−2.92 to 6.88) | 2.335 (−1.19 to 5.86) | −0.663 (−2.79 to 1.47) | −0.426 (−3.73 to 2.88) | 0.726 (−1.23 to 2.68) | 0.354 (−0.94 to 1.65) | 1.135 (−0.55 to 2.82) |
| | Up to HSC | 1.258 (−1.89 to 4.40) | 0.88 (−1.49 to 3.25) | 1.22 (−.57 to 3.01) | −0.215 (−2.58 to 2.15) | −0.826 (−2.16 to 0.51) | 0.085 (−1.03 to 1.20) | −0.12 (−1.40 to 1.16) |
| | Honors | 2.434 (−.85 to 5.72) | 3.485** (.77 to 6.20) | 1.237 (−.83 to 3.30) | −1.751 (−4.69 to 1.18) | −1.433 (−3.16 to 0.29) | −0.513 (−1.78 to 0.75) | −0.299 (−1.78 to 1.18) |
| | MS and above | 2.098 (−.87 to 5.08) | 2.477 (−.40 to 5.35) | 0.829 (−1.22 to 2.87) | 0.236 (−2.65 to 3.13) | −1.843 (−3.69 to 0.01) | −0.884 (−2.10 to 0.34) | −0.759 (−2.16 to 0.64) |
| | Illiterate | −4.199* (−8.06 to −0.34) | −1.665 (−5.17 to 1.84) | −0.513 (−2.69 to 1.66) | 2.749 (−0.21 to 5.71) | 2.401** (0.70 to 4.11) | 1.604* (.05 to 3.16) | 1.735* (0.01 to 3.46) |

(*Continued*)

**Table 4.** (*Continued*)

| Variables | | Anxiety MD (95% CI) | RD MD (95% CI) | Hygiene MD (95% CI) | Visibility MD (95% CI) | Treatability MD (95% CI) | PE MD (95% CI) | Recovery MD (95% CI) |
|---|---|---|---|---|---|---|---|---|
| Up to HSC | Up to primary | 0.718 | 1.455 | −1.883 | −0.212 | 1.552 | 0.269 | 1.255 |
| | | (−4.31 to 5.74) | (−2.15 to 5.06) | (−4.02 to 0.28) | (−3.34 to 2.91) | (−0.44 to 3.54) | (−1.09 to 1.62) | (−0.49 to 3.00) |
| | Up to SSC | −1.258 | −0.88 | −1.22 | 0.215 | 0.826 | −0.085 | 0.12 |
| | | (−4.40 to 1.89) | (−3.25 to 1.49) | (−3.01 to 0.57) | (−2.15 to 2.58) | (−0.51 to 2.16) | (−1.20 to 1.03) | (−1.16 to 1.40) |
| | Honors | 1.176 | 2.605 | 0.017 | −1.537 | −0.606 | −0.598 | −0.179 |
| | | (−2.31 to 4.66) | (−.22 to 5.43) | (−2.06 to 2.09) | (−4.26 to 1.18) | (−2.37 to 1.16) | (−1.92 to 0.73) | (−1.72 to 1.37) |
| | MS and above | 0.84 | 1.597 | −0.391 | 0.451 | −1.017 | −0.969 | −0.638 |
| | | (−2.34 to 4.02) | (−1.38 to 4.57) | (−2.44 to 1.66) | (−2.22 to 3.12) | (−2.90 to 0.87) | (−2.25 to 0.32) | (−2.11 to 0.83) |
| | Illiterate | −5.457** | −2.545 | −1.733 | 2.963* | 3.227** | 1.519 | 1.855* |
| | | (−9.47 to −1.44) | (−6.12 to 1.03) | (−3.92 to 0.45) | (0.20 to 5.72) | (1.48 to 4.97) | (−0.08 to 3.12) | (.08 to 3.63) |
| Honors | Up to primary | −0.458 | −1.15 | −1.9 | 1.325 | 2.158 | 0.867 | 1.433 |
| | | (−5.56 to 4.65) | (−4.97 to 2.67) | (−4.27 to 0.47) | (−2.23 to 4.88) | (−0.09 to 4.41) | (−0.61 to 2.34) | (−.45 to 3.31) |
| | Up to SSC | −2.434 | −3.485** | −1.237 | 1.751 | 1.433 | 0.513 | 0.299 |
| | | (−5.72 to 0.85) | (−6.20 to −0.77) | (−3.30 to 0.83) | (−1.18 to 4.69) | (−0.29 to 3.16) | (−0.75 to 1.78) | (−1.18 to 1.78) |
| | Up to HSC | −1.176 | −2.605 | −0.017 | 1.537 | 0.606 | 0.598 | 0.179 |
| | | (−4.66 to 2.31) | (−5.43 to 0.22) | (−2.09 to 2.06) | (−1.18 to 4.26) | (−1.16 to 2.37) | (−0.73 to 1.92) | (−1.37 to 1.72) |
| | MS and above | −0.336 | −1.008 | −0.408 | 1.987 | −0.411 | −0.371 | −0.46 |
| | | (−3.66 to 2.98) | (−4.26 to 2.24) | (−2.70 to 1.88) | (−1.19 to 5.17) | (−2.57 to 1.75) | (−1.79 to 1.05) | (−2.10 to 1.18) |
| | Illiterate | −6.633** | −5.150** | −1.75 | 4.500** | 3.833** | 2.117** | 2.033* |
| | | (−10.74 to −2.52) | (−8.94 to −1.36) | (−4.15 to 0.65) | (1.27 to 7.73) | (1.80 to 5.86) | (0.42 to 3.81) | (.13 to 3.94) |
| MS and above | Up to primary | −0.122 | −0.142 | −1.492 | −0.662 | 2.569* | 1.237 | 1.893* |
| | | (−5.04 to 4.80) | (−4.07 to 3.78) | (−3.84 to 0.85) | (−4.18 to 2.85) | (0.23 to 4.91) | (−0.20 to 2.68) | (.07 to 3.72) |
| | Up to SSC | −2.098 | −2.477 | −0.829 | −0.236 | 1.843 | 0.884 | 0.759 |
| | | (−5.06 to 0.87) | (−5.35 to 0.40) | (−2.87 to 1.2 2) | (−3.13 to 2.65) | (−0.01 to 3.69) | (−0.34 to 2.10) | (−0.64 to 2,16) |
| | Up to HSC | −0.84 | −1.597 | 0.391 | −0.451 | 1.017 | 0.969 | 0.638 |
| | | (−4.02 to 2.34) | (−4.57 to 1.38) | (−1.66 to 2.44) | (−3.12 to 2.22) | (−0.87 to 2.90) | (−0.32 to 2.25) | (−0.83 to 2.11) |
| | Honors | 0.336 | 1.008 | 0.408 | −1.987 | 0.411 | 0.371 | 0.46 |
| | | (−2.98 to 3.66) | (−2.24 to 4.26) | (−1.88 to 2.70) | (−5.17 to 1.19) | (−1.75 to 2.57) | (−1.05 to 1.79) | (−1.18 to 2.10) |
| | Illiterate | −6.297** | −4.142** | −1.342 | 2.513 | 4.244** | 2.487** | 2.493** |
| | | (−10.18 to −2.41) | (−8.03 to −0.26) | (−3.72 to 1.04) | (−0.68 to 5.70) | (2.12 to 6.37) | (0.82 to 4.16) | (0.64 to 4.34) |

(*Continued*)

**Table 4.** (Continued)

| Variables | Anxiety MD (95% CI) | RD MD (95% CI) | Hygiene MD (95% CI) | Visibility MD (95% CI) | Treatability MD (95% CI) | PE MD (95% CI) | Recovery MD (95% CI) |
|---|---|---|---|---|---|---|---|
| Illiterate | | | | | | | |
| Up to primary | 6.175* (.76 to 11.59) | 4 (−.34 to 8.34) | −0.15 (−2.60 to 2.30) | −3.175 (−6.73 to 0.38) | −1.675 (−3.89 to 0.54) | −1.25 (−2.97 to 0.47) | −0.6 (−2.65 to 1.45) |
| Up to SSC | 4.199* (.34 to 8.06) | 1.665 (−1.84 to 5.17) | 0.513 (−1.66 to 2.69) | −2.749 (−5.71 to 0.21) | −2.401** (−4.11 to −0.70) | −1.604* (−3.16 to −0.05) | −1.735* (−3.46 to 0.01) |
| Up to HSC | 5.457** (1.44 to 9.47) | 2.545 (−1.03 to 6.12) | 1.733 (−0.45 to 3.92) | −2.963* (−5.72 to −0.20) | −3.227** (−4.97 to −1.48) | −1.519 (−3.12 to 0.08) | −1.855* (−3.63 to −0.08) |
| Honors | 6.633** (2.52 to 10.74) | 5.150** (1.36 to 8.94) | 1.75 (−0.65 to 4.15) | −4.500** (−7.73 to −1.27) | −3.833** (−5.86 to −1.80) | −2.117** (−3.81 to −0.42) | −2.033* (−3.94 to −0.13) |
| MS and above | 6.297** (2.41 to 10.18) | 4.142* (.26 to 8.03) | 1.342 (−1.04 to 3.72) | −2.513 (−5.70 to 0.68) | −4.244** (−6.37 to −2.12) | −2.487** (−4.16 to −0.82) | −2.493** (−4.34 to −0.64) |

Abbreviations: CI, confidence interval; MD, mean difference; PE, professional efficiency; RD, relationship disruption.
*p-value<0.05.
**p-value<0.01.

difference in relation to stigma toward mental illness. Social and cultural factors may have contributed to the counterintuitive results. For instance, gender-based stereotypes often portray females as emotional and sensitive, which may eventually lead them to perceive mental illness as a natural part of being a woman, therefore, leading to develop internalized stigma. This narrative is more prevalent in Asian societies including in Bangladesh where gender-based norms and roles are predominantly defined by males. Furthermore, recent evidence suggests that females tend to show higher willingness for disclosure than males (He et al., 2021). While this may seem as a significant progress in terms of females being more vocal over mental health issues and seeking care, in some societies, this may pose risks for females to be more exposed to potential stigma. Besides, there is a general remark that females are more likely to be diagnosed with certain mental health conditions (e.g., depression and anxiety; Patel et al., 2006), which are often stigmatized. These accounts are particularly applicable in the context of Bangladesh, where females are expected to be more tolerant irrespective of the physical and mental health issues. The lack of awareness and exposure to mental health literacy including the inadequacy of mental health care may also contribute to the higher degree of stigma toward mental illness among females. Future qualitative or mixed-method studies should explore the sociocultural factors revolving around why females hold more stigma toward mental illness than males.

The study also reported a difference in geographical location when it comes to mental illness stigma. Participants residing in Dhaka reported more stigma for subscales hygiene, visibility, treatability, and recovery than those living in the rural areas. However, participants outside of Dhaka reported more stigma in terms of relationship difficulties than participants in Dhaka.

Growing research has demonstrated that urbanization and mental health problems are intricately linked significantly affecting social, economic, and environmental factors (Ventriglio et al., 2021). Evidence also showed that common mental health problems are higher in urban areas with causal factors identified as social disparities, social insecurity, pollution, and the lack of contact with people and nature (Trivedi et al., 2008; Srivastava, 2009; Ventriglio et al., 2021). The increasing number of people and greater exposure to mental illness may lead to increased stigma (Link et al., 1999) suggesting that people living in urban areas tend to have significantly higher levels of stigmatized attitudes toward mental illness (Girma et al., 2013; Zhang et al., 2019). Urban residents are more likely to experience unfair treatment from friends, law enforcers (e.g., police), difficulty sticking with a job, and unsafe environment (Forthal et al., 2019) that may reinforce stigma toward mental illness. Besides, with increased urbanization and greater social isolation, people may find it easy to distance themselves from those with mental health issues. Stigma in the form of relationship disruption was reported more by participants outside of Dhaka than participants residing in Dhaka. Social relationships in rural areas in Bangladesh are equipped with greater social ties and build with increased cooperation. Social ties may be jeopardized with the presence of mental health issues which is often believed to be caused by evil spirits and therefore, considered as untreatable. Besides, mental illnesses are also perceived as infectious in rural areas. Hence, avoiding people with mental health issues as seen in other Asian countries (Zhang et al., 2020) may be found beneficial with little insight into the disruption of social bonding. Future studies are required to fully understand the factors contributing to the geographical difference in the experience of mental illness stigma.

**Table 5.** Multiple regression analysis of sociodemographic variables and subscales of mental illness stigma scale (*n* = 325)

| Variables | Anxiety β (95% CI) | RD β (95% CI) | Hygiene β (95% CI) | Visibility β (95% CI) | Treatability β (95% CI) | PE β (95% CI) | Recovery β (95% CI) |
|---|---|---|---|---|---|---|---|
| **Age** | | | | | | | |
| <25 | Ref. | Ref. | Ref. | Ref. | Ref. | Ref. | Ref. |
| 25–40 | 2.78 | 1.45 | 0.55 | −0.86 | −1.02 | 0.02 | −0.25 |
| | (−0.16 to 5.73) | (−0.92 to 3.83) | (−1.10 to 2.21) | (−3.12 to 1.42) | (−2.40 to 0.35) | (−1.03 to 1.07) | (−1.49 to 0.99) |
| >40 | 4.64** | 2.52 | 0.42 | −1.53 | 0.23 | 0.54 | 0.82 |
| | (1.33 to 7.96) | (−0.15 to 5.19) | (−1.45 to 2.28) | (−4.09 to 1.03) | (−1.31 to 1.78) | (−0.64 to 1.73) | (−0.57 to 2.22) |
| **Location** | | | | | | | |
| Dhaka | Ref. | Ref. | Ref. | Ref. | Ref. | Ref. | Ref. |
| Outside Dhaka | 0.25 | 1.79** | −0.78 | −2.47** | −0.68 | −0.50 | −0.93** |
| | (−1.26 to 1.76) | (0.57 to 3.00) | (−1.63 to 0.06) | (−3.64 to −1.31) | (−1.38 to 0.03) | (−1.04 to 0.03) | (−1.57 to −0.30) |
| **Gender** | | | | | | | |
| Male | Ref. | Ref. | Ref. | Ref. | Ref. | Ref. | Ref. |
| Female | 2.21* | 2.22* | 1.44* | −0.98 | 0.95 | 0.40 | 0.22 |
| | (0.05 to 4.36) | (0.49 to 3.96) | (0.23 to 2.65) | (−2.64 to 0.69) | (−0.06 to 1.95) | (−0.37 to 1.17) | (−0.69 to 1.12) |
| **Socioeconomic status (SES)** | | | | | | | |
| Lower SES | Ref. | Ref. | Ref. | Ref. | Ref. | Ref. | Ref. |
| Lower–middle SES | −0.75 | −1.30 | −0.08 | 0.07 | 1.61** | 0.31 | 0.59 |
| | (−3.06 to 1.56) | (−3.16 to 0.56) | (−1.38 to 1.22) | (−1.71 to 1.86) | (0.53 to 2.68) | (−0.51 to 1.14) | (−0.39 to 1.60) |
| Middle SES | −2.16 | −2.58* | −2.31** | 0.10 | −0.20 | 0.11 | −0.37 |
| | (−4.63 to 0.32) | (−4.57 to −0.58) | (−3.70 to −0.92) | (−1.81 to 2.01) | (−1.35 to 0.96) | (−0.77 to 0.99) | (−1.41 to 0.67) |
| Higher SES | 0.94 | −2.02 | −1.36 | −1.54 | 0.49 | 0.21 | −0.75 |
| | (−3.35 to 5.23) | (−5.50 to 1.44) | (−3.77 to 1.06) | (−4.85 to 1.78) | (−1.51 to 2.49) | (−1.32 to 1.74) | (−2.56 to 1.06) |
| **Occupation** | | | | | | | |
| Service holder | Ref. | Ref. | Ref. | Ref. | Ref. | Ref. | Ref. |
| Student | 1.17 | 0.25 | 0.04 | <0.01 | −0.20 | −0.08 | 0.77 |
| | (−2.43 to 4.77) | (−2.65 to 3.15) | (−1.99 to 2.07) | (−2.78 to 2.78) | (−1.88 to 1.48) | (−1.36 to 1.21) | (−0.74 to 2.29) |
| Businessperson | 0.70 | 0.31 | 0.36 | 0.09 | 0.48 | 0.88 | 0.69 |
| | (−1.82 to 3.21) | (−1.72 to 2.34) | (−1.05 to 1.78) | (−1.86 to 2.03) | (−0.69 to 1.65) | (−0.02 to 1.78) | (−0.38 to 1.75) |
| Housewife | −0.26 | −1.39 | −0.37 | −0.25 | 0.09 | 0.03 | 0.13 |
| | (−3.47 to 2.95) | (−3.98 to 1.21) | (−2.17 to 1.44) | (−2.73 to 2.23) | (−1.40 to 1.59) | (−1.12 to 1.17) | (−1.23 to 1.48) |
| Unemployed | −1.42 | −2.47* | −0.02 | <0.01 | 0.69 | 0.20 | 0.17 |
| | (−4.31 to 1.46) | (−4.80 to −0.15) | (−1.64 to 1.61) | (−2.24 to 2.21) | (−0.66 to 2.03) | (−0.83 to 1.23) | (−1.05 to 1.38) |
| **Literacy** | | | | | | | |
| Up to primary | Ref. | Ref. | Ref. | Ref. | Ref. | Ref. | Ref. |
| SSC | 3.09* | 2.52* | −0.31 | −0.79 | 0.28 | −0.02 | 1.02 |
| | (0.39 to 5.80) | (0.33 to 4.70) | (−1.83 to 1.21) | (−2.89 to 1.30) | (−0.98 to 1.54) | (−0.98 to 0.95) | (−0.12 to 2.16) |
| HSC | 3.45* | 2.94* | −0.79 | −1.01 | 1.25 | −0.03 | 1.38* |
| | (0.54 to 6.36) | (0.59 to 5.29) | (−2.43 to 0.85) | (−3.26 to 1.24) | (−0.10 to 2.61) | (−1.07 to 1.01) | (0.15 to 2.61) |
| Honors | 3.11 | 0.67 | −0.89 | −0.76 | 1.92* | 0.30 | 1.63* |
| | (−0.12 to 6.35) | (−1.93 to 3.28) | (−2.71 to 0.94) | (−3.26 to 1.74) | (0.41 to 3.43) | (−0.85 to 1.46) | (0.27 to 2.99) |
| Master's and above | 2.67 | 1.29 | −0.21 | −1.61 | 2.94** | 1.03 | 2.72** |
| | (−0.80 to 6.14) | (−1.50 to 4.09) | (−2.16 to 1.75) | (−4.29 to 1.07) | (1.32 to 4.56) | (−0.21 to 2.27) | (1.26 to 4.18) |

(*Continued*)

**Table 5.** (*Continued*)

| Variables | Anxiety β (95% CI) | RD β (95% CI) | Hygiene β (95% CI) | Visibility β (95% CI) | Treatability β (95% CI) | PE β (95% CI) | Recovery β (95% CI) |
|---|---|---|---|---|---|---|---|
| Illiterate | 3.56 | 1.19 | −0.15 | −1.32 | −0.93 | −0.68 | −0.27 |
| | (−0.34 to 7.47) | (−1.96 to 4.34) | (−2.35 to 2.05) | (−4.34 to 1.69) | (−2.75 to 0.89) | (−2.07 to 0.72) | (−1.91 to 1.38) |
| Religion | | | | | | | |
| Muslim | Ref. | Ref. | Ref. | Ref. | Ref. | Ref. | Ref. |
| Non-Muslim | −0.35 | −0.21 | 0.06 | −1.45 | 0.25 | −0.21 | 0.54 |
| | (−2.32 to 1.63) | (−1.81 to 1.38) | (−1.05 to 1.18) | (−2.98 to 0.07) | (−0.67 to 1.17) | (−0.92 to 0.50) | (−0.30 to 1.37) |
| Knowledge about mental health | | | | | | | |
| Yes | −1.92 | −0.29 | 1.18* | 2.87** | 1.98 | 1.46** | 0.87 |
| | (−3.99 to 0.15) | (−1.96 to 1.38) | (0.01 to 2.34) | (0.59 to 3.78) | (1.02 to 2.95) | (0.73 to 2.20) | (−0.01 to 1.74) |
| No | Ref. | Ref. | Ref. | Ref. | Ref. | Ref. | Ref. |
| Presence of mental illness among family members | | | | | | | |
| Yes | 0.35 | 1.66 | −2.53 | 2.87 | −1.70 | 1.43 | 0.57 |
| | (−4.67 to 5.37) | (−2.38 to 5.71) | (−5.36 to 0.29) | (−1.00 to 6.74) | (−4.04 to 0.63) | (−0.36 to 3.22) | (−1.55 to 2.68) |
| No | Ref. | Ref. | Ref. | Ref. | Ref. | Ref. | Ref. |
| Treatment sought for mental health illness | | | | | | | |
| Yes | −1.78 | 0.61 | −0.69 | −0.42 | −0.64 | −1.85** | 0.07 |
| | (−5.51 to 1.96) | (−2.39 to 3.62) | (−2.79 to 1.41) | (−3.30 to 2.47) | (−2.38 to 1.10) | (−3.18 to −0.52) | (−1.50 to 1.64) |
| No | Ref. | Ref. | Ref. | Ref. | Ref. | Ref. | Ref. |

Abbreviations: β, co-efficient; CI, confidence interval; PE, professional efficiency; RD, relationship disruption.
*$p$-value<0.05.
**$p$-value<0.01.

The results showed that anxiety, relationship difficulties, and visibility significantly differed by age. Participants aged <25 years reported less stigma related to anxiety, relationship difficulties, and visibility than participants aged >40 years. Similarly, participants aged >40 years reported more stigma in relation to anxiety and relationship difficulties than participants aged between 25 years and 40 years. Previous research showed that mental illness stigma is associated with age. For example, in a recent study in the UK (Bradbury, 2020), people aged between 16 years and 18 years were found to have more stigmatized views than people aged between 40 years and over. Data also suggest that children under the age of 10 years demonstrated stigma with commonly used terms 'crazy', 'mad', and 'losing your mind' (Wilson et al., 2000). However, a study conducted in Singapore showed that there was an association between mental illness stigma, predominantly negative attitude toward people with mental illness and greater age (Chong et al., 2007). The study also noted that people aged between 65 years and 69 years and having higher stigma scores reported difficulty talking to people with mental illness blaming them for their conditions. Another study conducted among Asian men also showed that with increasing age stigma was found to be greater (Livingston et al., 2018). The reason for the age difference may lie in that young people have more conformity to social, familial, and peer pressure to behave and appear in such a way that is endorsed (Bradbury, 2020). On the other hand, people in the 40 years and above are likely to come to terms with individuals experiencing mental health problems contributing to the lower degree of stigma (Griffiths et al., 2014). However, another study conducted by Min (2019),

showed that people (Whites and Hispanics) aged between 55 years and over demonstrated more stigma. The findings of the present study provide an inclusive result that participants aged >40 years tend to have more stigmatized attitude than young participants. The potential reasons may include a generation gap around the beliefs about mental health and care as older adults may have grown up in a time when mental health remained poorly understood, therefore, stigmatized (Sirey et al., 2001). Additionally, older adults may emphasize physical health including comorbidities (e.g., diabetes, heart diseases, etc.) over mental health that can perpetuate stigmatized attitudes toward mental illness (Karel et al., 2012).

Socioeconomic divisions were found to have differed by subscale anxiety, relationship disruption, hygiene, treatability, professional efficiency, and recovery. Participants of middle SES reported more stigma related to anxiety, relationship disruption, hygiene, and treatability than participants with lower SES. Participants belonging to lower–middle SES tended to have more stigma regarding treatability, professional efficiency, and recovery than lower SES.

SES has been shown to be associated with mental illness stigma (Wang et al., 2021; Foster and O'Mealey, 2022). The results of the study suggested that participants with middle SES tend to have more stigma as evident in other Asian countries such as Pakistan, India, and China (Knifton, 2012). The potential reasons may be associated with a greater emphasis on social norms and conformity that raise concern about how mental illness is conceptualized by other people. In addition, a lack of exposure to people with mental illness may also contribute to greater stigma among middle socioeconomic groups. In contrast, lower SES groups may have more

exposure to people with mental health illness due to the greater prevalence of mental illness in these groups (Zhang et al., 2022). Greater exposure may lead to increased familiarity, empathy and eventually less stigmatization.

Participants with no literacy at all reported more stigma related to anxiety than participants with education level up to primary, HSC, Honors, and MS and above. Illiterate participants had more stigma related to relationship disruption than participants education level up to primary, Honors, and MS and above. Participants at the Honors stage of their education level had more stigma about visibility than illiterate participants. Similarly, participants with up to secondary, HSC, Honors, and MS and above reported more stigma pertaining to treatability than illiterate participants. Stigma related to professional efficiency was varied in terms of education levels with participants of HSC, Honors, and MS and above reported more than illiterate participants. Participants with HSC, Honors, and MS and above education levels had more stigma attached to recovery than illiterate participants.

Illiteracy has been found to be associated with greater levels of mental illness stigma. On the contrary, higher educational level is associated with lower levels of stigma in Asian countries such as Singapore and Korea and vice versa (Chong et al., 2007; Park et al., 2015; Jang et al., 2018). Lack of education and knowledge about mental health and illness can reinforce stereotypes and misconceptions. Illiterate people may have different ideas about the development, onset, and maintenance of mental illnesses including a firm believer in supernatural entities as contributing factors. The results showed that stigma varied in terms of subscales and education levels. For example, visibility and professional efficiency were reported more between participants with at least HSC level of education and illiterate people. This implies that mental illness stigma affect people across various education levels. Qualitative investigation is required why people with education tended to report stigma related to visibility and professional efficiency irrespective of levels and fields of study.

Participants who had knowledge about mental health issues had significant differences in the subscale's anxiety, visibility, treatability, professional efficiency, and recovery than those with no prior mental health issues. Participants having presence of mental illness among family members significantly differed in terms of hygiene than those with no presence of mental illness in the family. Finally, those who sought treatment for mental health illness had significant difference in the professional efficiency than those who did not.

People with and without prior knowledge about mental health and illness may have different implications. Research has found that prior knowledge and experience with mental health illness tend to have lower levels of stigma compared to those without prior knowledge (Milin et al., 2016; Hartini et al., 2018; Gulliver et al., 2019). Reduced levels of fear and prejudice toward those with mental illness can be attributed to the greater understanding and familiarity with mental illness. On the other hand, endorsing stigma toward people with mental illness is reinforced when there is a little contact (Holmes et al., 1999; Corrigan et al., 2001) or insufficient knowledge about it. These findings suggest that education and individual experiences can plan a crucial role in reducing mental illness stigma.

Presence of a family member with mental illness may offer an opportunity for increased education and understanding toward mental illness. This has the potential to reduce mental illness stigma at the personal level and can therefore, be used to develop community-based intervention involving family members. It is reasonable to assume that those with increased knowledge and understanding about mental illness are likely to seek mental health care. Conversely, lack of awareness or understanding may lead to underreporting of symptoms, delay in seeking treatment, and poor treatment outcomes (Hanlon et al., 2016). Therefore, it is essential to continue promoting mental health awareness and education to ensure people with mental illness receive the required care.

## Strengths and limitations

Amid the dearth of evidence pertaining to the prevalence of mental illness stigma and the factors associated with it, this investigated the mental illness stigma among people living in rural and urban areas with a number of important sociodemographic variables. The study suggests that the variables considered in the study should be explored in detail and be studied in conjunction with other relevant variables (e.g., the inclusion of deep-rooted cultural beliefs). The study also highlights the need for qualitative studies aiming to uncover the cultural beliefs in both rural and urban areas and the extent to which they predict stigma in terms of nature and strengths. The results of the study would be useful in designing appropriate mental health interventions in both contexts. Research should also explore stigma and attitudes related to suicidal behavior and help-seeking. The results will contribute to the development of culturally tailored mental health and suicide prevention strategies contributing to reducing stigma.

The authors acknowledge a few limitations. First of all, the nature of the cross-sectional study, in which causal inference between the variables and the central construct (stigma in this case), is not necessarily established. Generalization of the results can, therefore, be restricted. Self-report data may be prone to response bias. Additionally, the study employed a limited sample size with no rigorous sample-to-population comparison except for the employment of systematic random sampling. Collecting data from both targeted population as well as the total population and conducting statistical tests to identify discrepancies is recommended in future studies. Cronbach's alphas for four subscales were found to be moderately acceptable. In some cases, the relationship between items in a subscale might be influenced by specific participant characteristics or contexts, resulting in moderate alpha coefficients. In addition, the length of the subscales can impact the internal consistency. For example, shorter subscales with fewer items may lead to moderate alpha values. Finally, cultural variations in understanding and responding to the items within a subscale can affect the internal consistency reliability. The present study used a translated version of the measure, which was translated following the suggested procedures and pretested prior to the data collection. However, the use of culturally adapted and validated measures in future studies is strongly recommended.

## Conclusion

The results showed that mental health stigma is widespread in rural and urban settings of Bangladesh. The study contributes to one potential explanation of why a huge treatment gap (more than 95%) still exists in the country as reported by the recent most national mental health survey (World Health Organization, 2019). Age, SES, and education level can act as predictors of stigma toward mental illnesses. The results will be useful in developing mental health intervention programs (e.g., age-specific mental health awareness programs) both in rural and urban areas in Bangladesh to address

such widespread stigma (while also improving the adequacy and availability of existing mental health care).

**Open peer review.** To view the open peer review materials for this article, please visit http://doi.org/10.1017/gmh.2023.56.

**Data availability statement.** The data will be provided upon request to the corresponding author.

**Author contribution.** M.O.F., K.U.A.U., S.J., and D.C.S. contributed to the conceptualization and data collection. M.O.F. and A.H.K. contributed to the formal analysis, data curation, writing, and editing of the manuscript. E.C. and M.T.H. contributed to the reviewing and editing of the manuscript.

**Financial support.** The study was conducted by the Nasirullah Psychotherapy Unit under the project 'Community-based Mental Health Project' led by ADD International Bangladesh in association with Innovation for Wellbeing Foundation (IWF), and Disabled Child Foundation (DCF) and funded by Comic Relief, UK. The Funding ID is 4,318,758.

**Competing interest.** The authors declare no potential conflicts of interest with respect to the research, authorship, and/or publication of this article.

**Ethics statement.** The study was performed in line with the principles of the declaration of Helsinki. The research protocol was reviewed and approved by the Ethical Review Committee at the Department of Clinical Psychology, University of Dhaka Bangladesh (Project Number: IR201101). Both verbal and written informed consent were obtained from all participants included in the study. The participants gave consent to the submission of the study.

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
