## [Reviewer Report]

April 18, 2023

Professor Gary Belkin,

Editor-in-Chief, Global Mental Health

Dear Professor Gary Belkin,

Re: Mental Illness Stigma in Bangladesh: Findings from a Cross-sectional Survey

We would like to submit this manuscript for publication in Global Mental Health.

The manuscript highlights mental illness stigma among people living in rural and urban areas of Bangladesh. The manuscript highlights the gender difference and associated factors of mental illness stigma. The manuscripts would be of particular interest to the readers of the journal to have an extended insight into the prevailing stigma related to mental illness in Bangladesh and would inform a shared understanding of how mental illness stigma can be addressed in different contexts to achieve the principles of global mental health.

We also hope it will remind policymakers in government and non-governmental organizations to develop policies to develop an anti-stigma campaign for people in Bangladesh.

I will keep the co-authors updated about any changes and progress notified by the journal. This work was carried out with financial assistance provided by Comic Relief and there are no conflicts of interest. The manuscript has not been submitted elsewhere, is not under consideration by any journal. We have jointly approved the manuscript and agreed to its submission to Global Mental Health. We appreciate and accept the relevant copyright and other conditions set by the journal.

We look forward to hearing further from you.

Yours sincerely,

Md. Omar Faruk

Department of Clinical Psychology

University of Dhaka

Bangladesh

Email: orhaanfaruk07@gmail.com

---

## [Reviewer Report]

Having data on stigma from LMICs is essential and far too rare. This study provides interesting data from over 300 rural or urban-dwelling individuals in Bangladesh. However, at this point, there are a few issues that decrease the potential contribution of the paper.

First, the balance of the paper in terms of a review of past research, presentation of details on methods, and findings need to be reconsidered. Specifically, the front end of the paper could be significantly streamlined. Much of what is provided here is well-known and can be condensed. Further, rather than justifying the study of stigma or the impact of stigma, the front end might be stronger if it focused on how what is generally known is in line with or is different from what might be expected in Bangladesh.

Second, more details on the methods need to be provided. Was this a convenience sample? How were they recruited? Why did they use the Day scale which was developed and tested on college students rather than other more well-known and routinely used scales, many from the authors that you cite? Do you need to show the ANOVA findings when the regression is also reported?

Third, is this the first study of its kind? Bangladesh was included in the Stigma in Global Context Study which used a representative sample of the country and several different scales including social distance, treatment-based stigma, etc. While there was no paper located that looked specifically at Bangladesh alone, there are a few publications in which Bangladesh was included. The one piece in the American Journal of Public Health could be quite useful here because it highlights the importance of doing more research on Bangladesh. This is because Bangladesh is among the top countries in terms of stigma levels. There was also a paper on stigma in Eastern vs Western countries which also seems to support this paper’s goals.

---

## [Reviewer Report]

The findings make an important contribution to the literature considering that it is the first study of its kind in Bangladesh. Below are comments for ways to improve the impact of the manuscript.

The introduction is a bit disorganized. The most important task of the introduction should be to establish what is and isn’t known about stigma in South Asia in general and in Bangladesh in particular. The first 3 paragraphs can be condensed considerably and then the authors can focus on cultural factors and the South Asian context. From what I can tell from the introduction, previous studies in Bangladesh have only examined the experience of stigma from the perspective of people with mental illness, so this would be the first study of the predictors of community stigma in Bangladesh. Although this is stated in the conclusion, the authors should state this more clearly in the introduction as it helps establish the study’s significance.

The door to door recruitment method is impressive and the statement that there were no refusals is also impressive. The lack of refusals calls for some explanation since it would be unheard of in Europe or North America. It would also be helpful if there were a way to compare the characteristics of the sample to the characteristics of the Bangladesh population in general, and any differences between the two discussed.

In the discussion, since there have been no prior studies in Bangladesh it would be helpful if the authors could contextualize the findings by comparing them to what has been found in other locations in South Asia, especially Pakistan, and discuss what may account for any differences that might be noted.

---

## [Reviewer Report]

This is an interesting article that makes a contribution to the understanding of the phenomenon of stigma, which is global but with characteristics specific to each context and culture.

Recommendations for authors:

1. Provide more details about the representativeness of the sample obtained, how much is it similar to the source population?

2. Describe if there was a specific procedure for the diagnosis of mental disorder by the research assistants, was there a structured interview?

3.- The results of Cronbach’s Alpha for the total stigma scale are deficient (0.67), which introduces possible measurement errors. If the questionnaire has 7 dimensions and these are partially exclusive, it is preferable to provide the Alpha values for each dimension.

---

## [Reviewer Report]

The authors were very responsive to reviewer feedback and the manuscript is greatly improved. I only ask that they thoroughly proof-read it before submitting the final version as there are still some errors.

---

## [Reviewer Report]

The authors have responded to all the reviewers' comments and the revised version has included all the aspects that were requested.